# Health-Promoting Factors and Their Relationships with the Severity of Symptoms in Patients with Anxiety Disorders during the COVID-19 Pandemic

**DOI:** 10.3390/healthcare11081153

**Published:** 2023-04-17

**Authors:** Marcin Jarosz, Paweł Dębski, Patryk Główczyński, Karina Badura-Brzoza

**Affiliations:** 1Department of Psychiatry, Faculty of Medical Sciences in Zabrze, Medical University of Silesia, 40-000 Katowice, Poland; 2Institute of Psychology, Humanitas University in Sosnowiec, 34-112 Sosnowiec, Poland

**Keywords:** COVID-19 pandemic, anxiety disorders, prohealth behaviors, life satisfaction, anxiety, depression

## Abstract

Background: Anxiety disorders are one of the most common mental disorders in the modern world. The COVID-19 pandemic has led to the onset of many mental disorders in people who did not have them before. It can be suspected that in people who already had anxiety disorders before the pandemic, their quality of life has significantly deteriorated. Aim: The aim of the study was to assess the relationships between life satisfaction, acceptance of illness, the severity of anxiety and depression symptoms and health behaviors in a group of patients diagnosed with anxiety disorders during the COVID-19 pandemic. Material and methods: The study was conducted in the period from March 2020 to March 2022. There were 70 people among the respondents, including 44 women aged 44.06 ± 14.89 years and 26 men aged 40.84 ± 16.72 years. All persons were diagnosed with generalized anxiety disorder. Patients with other co-occurring disorders were excluded, i.e., depression and signs of organic damage to the central nervous system, as were those with cognitive disorders that prevented the completion of the questionnaires. The following scales were used in the study: the Satisfaction with Life Scale (SWLS), Acceptance of Illness Scale (AIS), Health Behavior Inventory (HBI) and Hospital Anxiety and Depression Scale (HADS). Spearman’s rank correlation coefficient and the Mann–Whitney U test were used for statistical analyses. Results: In the Satisfaction in Life questionnaire, the respondents obtained an average score of 17.59 ± 5.74 points. In the AIS scale, the mean score obtained by the patients was 27.10 ± 9.65 points. In the overall Health Behavior Inventory (HBI), the average score was 79.52 ± 15.24 points. In the HADS questionnaire, probants obtained an average of 8.17 ± 4.37 points in the depression subscale and 11.55 ± 4.46 points in the anxiety subscale. In addition, there were significant negative correlations between life satisfaction (SWLS) and the severity of anxiety and depression (HADS). The lower the perceived quality of life, the significantly higher the anxiety and depressive disorders. The result obtained in the Health Behavior Inventory (HBI) as well as in the subscale of Prohealth Activities (PHA) was negatively associated with the severity of anxiety symptoms. Prohealth activities should therefore be developed to prevent anxiety disorders, as well as to promote positive mental attitudes. In the study, the average result obtained in the subscale of positive mental attitudes correlated negatively with both anxiety and depressive symptoms. Conclusions: Life during the pandemic was assessed by patients as unsatisfactory. Health-promoting behaviors, and especially positive mental attitudes, may play a protective role in relation to anxiety and depressive symptoms in a situation of increased stress related to the COVID-19 pandemic in a group of patients with anxiety disorders.

## 1. Introduction

### 1.1. Life Satisfaction

Life satisfaction is a positive way of evaluating one’s life as a whole [1]. It is a distinct construct representing the cognitive and global evaluation of the quality of life (QoL) and is one of the components of subjective well-being (SWB) [2]. It can be seen that life satisfaction is related to the concept of quality of life, which is defined by the World Health Organization (WHO) as “an individual’s perception of his life position in the cultural context and value system in which he lives, and in relation to the tasks, expectations and standards set environmental conditions” [3]. Interest in the quality of life in medicine continues to grow [4]. Research conducted in this area gives the opportunity to improve the quality of medical care services [5]. Quality of life in medical terms is associated primarily with good health, which is understood as mental, physical and social well-being [6]. According to Lalonde’s concept, this well-being results from the coexistence of specific genetic, environmental, lifestyle and healthcare factors [7]. Our lifestyle consists of e.g., habitual patterns of behavior that are intended to maintain or restore health. These are the so-called prohealth behaviors, which are influenced by, among others, modeling behavior in the family, demographic and social conditions, personality traits and the impact of society and culture [8]. Life satisfaction, together with prohealth behaviors and social support, are health predictors [9,10,11]. At present, it is known that there is a negative correlation between life satisfaction and mental disorders [12], and mental disorders themselves are associated with higher mortality [13].

### 1.2. Impact of COVID-19 on Patients with Anxiety Disorder

The announcement of the COVID-19 pandemic in early 2020 was undoubtedly a turning point that had a huge impact on societies. People had to reorganize their functioning in private and professional life. The introduced restrictions and isolation had a negative impact on health behaviors and, thus, the physical and mental health of many people [14]. Another unfavorable factor for health was perceived stress. Concerns about one’s own health and the health of one’s relatives [15] might have become one of the generators of the increase in the prevalence of anxiety and depressive disorders [16,17,18], and also increased the risk of post-traumatic psychiatric symptoms [19]. The introduction of restrictions contributed to the deterioration of social functioning and the quality of life [20] and the restrictions related to the reduction of social support could have had an impact on social functioning itself, as well as on changes in health-related behaviors [21]. It seems that this has been a particularly difficult time for people who were already struggling with mental health problems before the pandemic [22]. People who were receiving treatment for previously diagnosed anxiety disorder had a higher incidence of adverse mental and behavioral health conditions during the pandemic, such as depressive and anxiety symptoms, abuse of psychoactive substances to cope with stress and consideration of suicide [23]. This is due to the fact that individuals with current psychiatric disorders experienced higher psychological distress due to less effective coping strategies and more difficult access to mental health professionals during the pandemic [24].

In relation to the observed increase in anxiety disorders since the announcement of the pandemic, research studies have begun to focus on its impact on mental health by examining vulnerability and resilience factors [25]. During this period, it was noticed that factors such as reduced healthy diet, reduced physical activity and more substance abuse were associated with higher levels of psychological strain [26]. Other factors that increased anxiety were frontline medical personnel, chronic diseases, presenting symptoms of SARS-CoV-2 infection or contact history [27], female gender, personality domains of negative affect, detachment [28] and increased social media exposure [29], household relationship difficulties, fear and uncertainty, external restrictions and lack of social support [30].

Based on the above research evidence, we have reason to speculate that the life satisfaction of people with anxiety disorders could have been affected during the COVID-19 pandemic. It is imported to look for protective factors, which can be modified by psychosocial intervention.

## 2. Aim

The main objective of the study was to assess the relationships between health-promoting factors such as life satisfaction, acceptance of illness and health behaviors and the severity of symptoms of anxiety and depression in a group of patients diagnosed with generalized anxiety disorders (GAD) during the COVID-19 pandemic. The aim of the study was also to assess gender differences in terms of the studied variables. Finding significant relationships between the above variables and the severity of psychopathological symptoms may contribute to the determination of additional treatment strategies for people with GAD, which could be adapted in the event of restrictions resulting from the epidemic situation. It will also allow for faster interventions in new patients diagnosed with GAD.

## 3. Material and Methods

The study was conducted in the period from March 2020 to March 2022. Patients received the questionnaires during a personal visit to the Mental Health Clinic (MHC), located in the Upper Silesian metropolis, Poland. Initially, 150 people who had been diagnosed with generalized anxiety disorder, had a medical history of at least one year and reported personally to MHC in the abovementioned study period qualified for the study. Questionnaires were returned by 89 patients, and only fully completed questionnaires were included in the final analysis. There were 70 people among the respondents, including 44 women aged 44.06 ± 14.89 years and 26 men aged 40.84 ± 16.72 years. All persons were diagnosed with generalized anxiety disorder. Patients with other co-occurring disorders were excluded, i.e., depression and signs of organic damage to the central nervous system, as were those with cognitive disorders that prevented the completion of the questionnaires. All respondents agreed to participate in the project. The sociodemographic characteristics of the respondents are presented in Table 1.

The following tools and psychometric questionnaires were used to assess the parameters studied:Original demographic data questionnaire containing, e.g., questions about gender, age, place of residence, relationships or level of education. In addition, patients were asked about their subjective assessment of stress, assessing its severity on a 10-point scale.Satisfaction with Life Scale (SWLS).

The scale by Diener, Emmons, Larson and Griffin in the Polish adaptation of Juczyński allowed for the study of life satisfaction, understood as a subjective assessment of the quality of functioning. It contains five items. The respondents were asked to respond to each of the statements by specifying to what extent each of them applied to his/her life so far, from “strongly agree” (7 points) to “strongly disagree” (1 point). The scores were summed up and the result determined the level of satisfaction with life. Scores ranged from 5 to 35 points. They were converted to standardized units on the sten scale. Scores in the range of 1–4 sten were considered low and in the range of 7–10 sten as high. Results in the 5–6 sten range corresponded to average values [14].

3.Hospital Anxiety and Depression Scale (HADS)

The scale consists of two independent subscales containing 7 statements each, one of which assesses the current severity of anxiety symptoms (HADS-A) and the other the severity of depression symptoms (HADS-D). Achieving 0–7 points in each of the subscales is considered the norm [15].

4.The Acceptance of Illness Scale (AIS)

The scale consists of 8 statements expressing specific difficulties and limitations caused by the disease, ranging from 8 to 40 points. The general measure of the degree of acceptance of the disease is the sum of the points obtained. Scores can be grouped into three score ranges: 8–19 (low), 20–35 (medium) and 36–40 (high). A low score means a lack of acceptance and adaptation to the disease and a strong sense of mental discomfort. A high score indicates acceptance of one’s own disease state and the lack of negative emotions related to the disease [16].

5.Health Behavior Inventory (HBI)

The questionnaire by Jurczyński is a tool for measuring health behaviors. It allows determining the general intensity of prohealth behaviors and its four subscales—Positive Mental Attitude (PMA) (avoiding strong emotions, tension, and stressful and depressing situations), Proper Eating Habits (PEH) (type and frequency of food consumed), Preventive Actions (PA) (following health recommendations, obtaining information about health and illness), and Prohealth Activities (PhA) (everyday habits: sleep, recreation, physical activity). This tool consists of 24 statements to which the respondent responds on a five-point scale, where one means “almost never” and five means “almost always”. The possible overall score is in the range of 24–120 points. The higher the result, the greater the intensity of the declared prohealth behaviors. This indicator, after being converted into standardized units, was interpreted in the sten scale. This test is the only tool in Polish cultural conditions that allows for a global assessment of health behaviors involving the most important spheres of prohealth and preventive behaviors [14].

The prepared form contained information for participants about the aim of the study and discussed the instructions for filling out the questionnaire for the tests used.

## 4. Statistical Analysis

Standard statistical procedures were used in the analyses. The Mann–Whitney U test was used to assess the significance of differences between the study groups. Spearman’s rank correlation coefficient was used to assess the relationships between the data. The significance level α ≤ 0.05 was assumed as statistically significant. Calculations were made in Statistica version 13.3 and Excel 2016.

## 5. Ethical Consideration

The Bioethics Committee of the Medical University of Silesia approved the study (PCN/0022/KBI/67/21).

Patients were informed about the anonymity and confidentiality of the research. What is more, they were informed that they could stop the study whenever they wanted. Information about the study and informed consent were included in the first part of the prepared form.

Patients or researchers were not offered any compensation as an incentive to participate. The authors received no specific funding for this study.

## 6. Results

The description of the study group is included in Table 1. In order to properly find the relationship between anxiety, depressive disorders, quality of life and prohealth behaviors, we used the following scales in the study: Satisfaction with Life scale (SWLS), Acceptance of Illness Scale (AIS), Health Behavior Inventory (HBI) and Hospital Anxiety and Depression Scale (HADS). Using these scales, it was possible not only to estimate the parameters of the subjects but also to compare them with other research studies. The scales used, along with the scores obtained in the study, are presented below.

### 6.1. Satisfaction with Life Scale (SWLS)

In the SWLS scale, responders were assessed in the range of 5–35 points. Ranges are divided into six groups: 30–35 points (extremely satisfied), 25–29 points (satisfied), 20–24 points (slightly satisfied), 15–19 points (slightly dissatisfied), 10–14 (dissatisfied) and 5–9 points (extremely dissatisfied).

Analyzing the results obtained in the study group in the SWLS, an average score of 17.59 ± 5.74 points was obtained, which, in terms of stens, allows estimating the level of life satisfaction as slightly dissatisfied (Table 2). After division into groups, 17.76 ± 6.05 points were obtained, respectively, for women and 17.40 ± 5.375 pts. for men; the difference was not statistically significant (Table 3).

### 6.2. Acceptance of Illness Scale (AIS)

Analyzing the results obtained in the study group on the AIS scale, an average score of 27.10 ± 9.6 points was obtained, indicating an average level of acceptance of the disease (Table 2). Analyzing the results separately in the group of women, 25.97 ± 9.08 points were obtained, and in the group of men, 29.28 ± 10.52 points were obtained; the difference was not statistically significant (Table 3). The total scores of AIS were between 8 and 40 points. The low AIS score indicates a lack of adjustment to the illness, no acceptance of the illness, and mental discomfort. The high score indicates good acceptance of the disease.

### 6.3. Health Behavior Inventory (HBI)

The HBI comprises 24 statements presenting health behaviors in four categories: Positive Mental Attitude (PMA), Proper Eating Habits (PEH), Preventive Actions (PA) and Prohealth Activities (PhA). Each category was scored on a five-point scale, where one means almost never, two—rarely, three—occasionally, four—often and five—almost always. The sum of the scores from the whole questionnaire forms the general indicator of the intensity of HBs. The raw scores were transformed into 1–10 sten, where 1–4 sten means low scores, 5–6 sten—average scores and 7–10 sten—high scores.

Analyzing the results obtained in the study group in HBI, an average score of 79.52 ± 15.24 points was obtained, and after division into groups, the women obtained 81.37 ± 13.61 points and the men obtained 76.24 ± 17.78 points. In terms of sten values, all three results are at the average level. The groups did not differ statistically in terms of the overall HBI score or in individual subscales (Table 2 and Table 3).

### 6.4. Hospital Anxiety and Depression Scale (HADS)

Analyzing the results obtained in the study group in the HADS scale, the mean score of 8.17 ± 4.37 points was obtained in the depression subscale and in the anxiety subscale, 11.55 ± 4.46 points were obtained (Table 2). After division into groups, in the depression scale, women obtained 8.23 ± 4.32 points and men obtained 7.92 ± 4.57 pts., and the difference was not statistically significant. In the anxiety subscale, 11.83 ± 4.55 points were obtained for women and 10.92 ± 4.26 pts. for men, and the difference was not statistically significant (Table 2 and Table 3). In both examined groups, the level of depression was estimated as mild and the symptoms of anxiety were estimated as moderate,

### 6.5. Analysis of Relationships between the Tested Parameters

The analysis of the relationship between the examined parameters was presented for all patients together. Significant negative correlations between satisfaction with life (SWLS) and severity of anxiety and depression (HADS) were noted. In addition, the overall severity of declared health behaviors (HBI) as well as the score of the subscale of Prohealth Activities (PHA) were negatively associated with the severity of anxiety symptoms. The average result obtained in the subscale of Positive Mental Attitude (PMA) correlated negatively with both anxiety and depressive symptoms (Table 4). We found no gender differences in our study. Based on these results, it can be assumed that regardless of gender, the pandemic caused a significant reduction in the quality of life. Moreover, prohealth values and their protective effects seem to be universal for both genders.

## 7. Discussion

In the study group of patients suffering from a generalized anxiety disorder, we assessed the relationship between the severity of anxiety–depressive symptoms presented by the respondents and their satisfaction with life, acceptance of the disease and prohealth behaviors undertaken by these people during the COVID-19 pandemic. The average duration of anxiety disorders in the study group was 5.95 years, and most of the respondents were women. This fact is consistent with the commonly known statistical data, which indicate that women are more likely to suffer from anxiety disorders [31]. In the conducted study, we found that the presence of anxiety and depressive symptoms had a negative relationship with the assessment of life satisfaction. Similar results were obtained by other researchers. In a meta-analysis of 23 studies comparing the quality of life of patients suffering from anxiety disorders and people from the control group without anxiety disorders, a significant decrease was found in all domains of quality of life, i.e., mental health, physical health, work, and functioning social and family life in the group of patients compared to healthy people [32]. The presence of anxiety disorders can have a negative impact on quality of life. However, it should be taken into account that our study was conducted in the particularly difficult period of the COVID-19 pandemic. The results obtained in the SWLS in the study group amounted to 17.59 ± 5.74 points, which, in terms of stens, allows us to estimate the level of life satisfaction as low. Similarly, in a study by Passos et al., respondents rated their satisfaction with life on the SWLS scale slightly below average (median 18). In the abovementioned study, as well as in our analysis, the presence of depressive symptoms correlated negatively with life satisfaction [33]. Factors affecting the assessment of the quality of life during a pandemic may include restrictions related to preventing the spread of coronavirus.

Gonzalez-Bernal et al. assessed life satisfaction in people during the restrictions caused by the pandemic. They showed that the SWLS score was negatively correlated with the number of days confined at home and the number of people who lived together while positively correlated with the number of rooms in the home and the number of children under 18 years of age. In addition, other factors that were positively associated with the quality of life were indicated, such as employment, access to private outdoor space, the feeling of receiving enough information and the lack of imposed isolation, as well as male gender [34]. In our study, gender did not significantly differentiate the SWLS score. The above research shows what impact the pandemic had on the quality of life. However, the people included in these studies had no previously diagnosed anxiety disorders. In our study, the average HADS-A score was 11.55 ± 4.46 points and HADS-D 8.17 ± 4.37. These results are slightly higher compared to the data obtained in the White and Van Der Boor study, where the average score for the HADS anxiety subscale was 10.23 ± 4.98 points and for the depression subscale, 7.57 ± 4.39 points. However, in the study mentioned above, 26.3% of respondents were under treatment for mental disorders, including 14.3% for anxiety disorders [35]. Much lower results were obtained in the study by Özdin and Özdin, where for anxiety and depression, respectively, 6.8 ± 4.2 points and 6.7 ± 4.2 pts were obtained. [36]. In this study, 21.8%, a minority, were people with a previously diagnosed mental disorder. In both of the above studies, women had significantly higher levels of anxiety compared to men, which was not shown in our study. This may be due to the fact that the study group included in our study was much smaller. It is also worth noting that in the study by Özdin and Özdin, the level of anxiety and depression was significantly higher in the group of people with a previously diagnosed mental disorder [37].The increase in anxiety and depression levels during this particularly difficult period is understandable. The outbreak of the pandemic was an unexpected phenomenon, which resulted in significant feelings of dread, including fear for health and financial stability. Uncertainty itself is a stress factor that can significantly affect the mental state of individuals [38]. Another important phenomenon related to uncertainty is intolerance of uncertainty (IU). It is a construct consisting of 4 dimensions: the person perceives uncertainty as a stressful phenomenon, it is an intolerable situation that should be avoided, and the person considers uncertainty as something unfair, which leads to the inability to take action [39]. Research shows that IU may play a key role in anxiety disorders. In the cognitive model, an individual with anxiety disorders will show intolerance to the state of uncertainty, will have a positive belief about worrying, will use cognitive avoidance and will present a negative orientation to the problem [40].

It can therefore be assumed that unpredictable events may significantly worsen the mental state of people with anxiety disorders and impair their quality of life. This was proved by Langhammer et al., which assessed the impact of the pandemic on patients with anxiety disorders. During the first wave of the pandemic, the symptoms of panic disorder, phobic symptoms and nonspecific anxiety significantly intensified in these patients [41]. However, in our study, based on the collected data, we could not determine whether psychopathological symptoms changed due to the pandemic. A variable that may also affect patients’ satisfaction with life and remains in a positive correlation with it is health behavior [42]. Meta-analyses indicate that a low level of health behaviors has a negative impact on mental health [43,44], and undertaking physical activity reduces the level of depression and anxiety [45]. The period of the pandemic and the introduction of restrictions resulted in a decline in health behaviors [44,45,46]. In our study, we found a negative correlation between the overall score obtained in the Health Behavior Inventory (HBI) and the severity of anxiety. It can be assumed that the improvement of health behaviors will have a positive impact on mental health. In particular, interventions should concern the area of ”health practices”. In the subscale of Prohealth Activities (PHA), we observed that it was negatively correlated with anxiety. Therefore, intervention in this area should be undertaken, e.g., in the form of psychoeducation, to motivate patients to undertake, for example, physical activity, ensure proper nutrition or follow medical recommendations. People with anxiety symptoms may benefit from activities aimed at reinforcing health behaviors [47].

Another issue concerns the Positive Mental Attitude (PMA) subscale, which was negatively correlated with depression and anxiety. Again, referring to the cognitive model, the lack of a positive attitude or having negative beliefs are characteristics of anxiety disorders and depression [48]. Using psychotherapy techniques, we can assume that as a result of cognitive restructuring in this area, we will reduce the symptoms of depression and anxiety and improve the quality of life of patients. Therefore, participation in psychotherapy, associated with building a positive mental attitude, can be an important factor in reducing the symptoms of anxiety and depression and increasing hope, optimism and satisfaction with life. The lack of correlation between the results of the AIS scale and the intensity of anxiety and depression is surprising. In a study evaluating the impact of rehabilitation after coronary artery bypass grafting (CABG) on the reduction of depression and anxiety symptoms, researchers found a strong negative correlation between the severity of these symptoms and the acceptance of the disease, which was measured with the AIS scale [49].

Areas assessed using this scale include low self-esteem related to the disease, a sense of burden for others and causing embarrassment among people with whom one spends time, a sense of lack of independence and the inability to do what one likes the most [50]. Thus, the AIS is used to assess areas related to self-acceptance, a sense of autonomy, the possibility of self-development and impact on the environment, and relationships with others. According to Riff’s model, these dimensions have a significant impact on mental well-being [51]. The study of the relationship between the acceptance of the disease and the severity of symptoms of anxiety and depression among people with mental disorders, in light of the results obtained, requires further exploration.

## 8. Study Limitations

The presented study, like any study based on a questionnaire, has some limitations. First of all, the data was collected in the form of self-report questionnaires—there was no control over the course of filling out the questionnaires. The limitation of the work is also the small number of respondents and the collection of research material within one disease entity, without creating comparative groups. Data from one center limits the possibility of generalizing the results.

Another limitation of the study was also that the patients were already in active treatment. We also did not report the type of treatment the patients were undergoing. An additional point is the lack of information about the impact of COVID-19 on income, loss of loved ones, illness, and work and the state of social isolation at the time of completing the questionnaire by patients. All these factors could weaken our conclusions.

Due to the length of the article, the study did not describe the significance of differences in the scope of the studied variables resulting in the study group from sociodemographic differences, although the authors are aware that they exist.

## 9. Conclusions

Patients with GAD assessed their quality of life at a slightly dissatisfied level during the pandemic. This may be due to both the presence of mental disorders and external factors. In the abovecited research, it was found that in the case of anxiety disorders, people suffering from them assess their quality of life as worse compared to people without anxiety disorders, and the quality of life was worse compared to the period before the outbreak of the pandemic. It is possible that both factors could have had a synergistic effect on the results obtained in the SWLS scale in our study.We can assume that the presence of psychopathological symptoms in the form of anxiety and depression had a negative impact on the quality of life. The intensification of the above symptoms could have been caused by emerging fears related to the pandemic and less effective mechanisms of coping with stress.Among the respondents, a negative correlation between prohealth behaviors and the severity of anxiety symptoms was found. We can therefore assume that the reduction of prohealth behaviors will have an adverse effect on the severity of psychopathological symptoms. This may especially apply to areas related to prohealth activities and positive mental attitudes. It can therefore be assumed that improving prohealth behaviors may reduce symptoms of anxiety and, thus, improve the quality of life. Appropriate psychoeducation-based interventions can be implemented, for example, during control visits.Considering the above results, working with patients in the area of health behaviors should be part of the therapy of GAD, especially in situations of increased and chronic stress (as in the case of a pandemic). Patients should be motivated to undertake physical activity, receive advice about where to look for information on healthy eating and get information about the benefits of reducing substance use. Another area of working with patients is a positive mental attitude. At this point, we can refer to resilience, or the ability to maintain a state of normal equilibrium in the face of extremely unfavorable circumstances [51]. The improvement of the resilience effect can be achieved by enhancing positive emotions, cognitive reappraisal, relaxation and psychoeducation [52].The results of this work can set the direction for further studies with an improved methodology and larger sample sizes. The results of studies looking for protective factors among people with mental disorders may contribute to the improvement of therapeutic methods, the use of which will be possible in the event of restrictions.

## Figures and Tables

**Table 1 healthcare-11-01153-t001:** Demographic characteristics of the sample.

Variable	Women (n = 44)	Men (n = 26)	Total (n = 70)
AGE (mean ± SD)	44.06 ± 14.89	40.84 ± 16.72	42.88 ± 15.23
**Marital Status—Total (%)**			
Single/Divorced	11 (26%)	14 (53.8%)	25 (36.8%)
In relationship	31 (74%)	11 (%)	42 (63.2%)
**Living with Someone—Total (%)**			
Yes	30 (71.4%)	17 (65.4%)	47 (70.1%)
No	12 (28.6%)	8 (34.6%)	20 (29.9%)
**Education Level—Total (%)**			
Tertiary	16 (38%)	5 (19.2%)	21 (31.3%)
Secondary (+students of universities)	20 (47.6%)	17 (65.3%)	37 (55.2%)
Middle school	6 (14.4%)	3 (15.5%)	9 (13.5%)
Duration of disease (AGE)	3.76	9.17	5.95
BMI	25.86	28.57	26.90
Is the current situation related to the spreading of the SARS-CoV-2 virus stressing?	5.14	4.56	4.93

**Table 2 healthcare-11-01153-t002:** Descriptive statistics of variables for the study group as a whole (N = 70).

Variable	Mean	St. dev.	Median	Min.	Max.	PU − 95%	PU + 95%
AIS	27.101	9.653	26.000	8.000	45.0000	8.268	11.600
SWLS	17.594	5.740	17.000	7.000	33.0000	4.917	6.898
HADS-A	11.551	4.464	12.000	2.000	20.0000	3.824	5.364
HADS-D	8.174	4.379	8.000	0.000	17.0000	3.751	5.262
HBI	79.522	15.241	81.000	19.000	111.0000	13.054	18.314
PEH	3.015	0.978	2.833	1.000	4.8333	0.836	1.178
PA	3.744	0.781	3.833	1.667	5.0000	0.668	0.942
PMA	3.353	0.717	3.333	1.833	5.0000	0.613	0.865
PHA	3.331	0.570	3.167	2.167	4.5000	0.487	0.687

AIS—Acceptance of Illness Scale, HADS-A—Hospital Anxiety and Depression Scale—Anxiety, HADS-D—Hospital Anxiety and Depression Scale—Depression, SWLS—The Satisfaction with Life Scale, HBI—Health Behavior Inventory, PMA—Positive Mental Attitude, PEH—Proper Eating Habits, PA—Preventive Actions, PHA—Prohealth Activities.

**Table 3 healthcare-11-01153-t003:** Gender differences in the studied variables.

	Women (n = 44)					Men (n = 26)				
Variable	Mean	St. Dev.	Min	Median	Max	Mean	St. Dev.	Min	Median	Max
AIS	25.977	9.088	8.000	25.000	40.000	29.280	10.522	10.000	31.000	45.000
SWLS	17.767	6.051	7.000	18.000	33.000	17.400	5.377	8.000	16.000	31.000
HADS-A	11.837	4.556	2.000	13.000	20.000	10.840	4.269	4.000	10.000	18.000
HADS-D	8.233	4.325	0.000	8.000	17.000	7.920	4.573	0.000	8.000	16.000
HBI	81.372	13.609	56.000	82.000	111.000	76.240	17.789	19.000	79.000	100.000
PEH	3.171	0.978	1.167	3.000	4.833	2.743	0.957	1.000	2.833	4.500
PA	3.869	0.737	2.167	4.000	5.000	3.535	0.841	1.667	3.667	5.000
PMA	3.286	0.756	1.833	3.333	4.833	3.438	0.642	2.167	3.417	5.000
PHA	3.306	0.590	2.167	3.167	4.500	3.389	0.549	2.500	3.417	4.333

AIS—Acceptance of Illness Scale, HADS-A—Hospital Anxiety and Depression Scale—Anxiety, HADS-D—Hospital Anxiety and Depression Scale—Depression, SWLS—The Satisfaction with Life Scale, HBI—Health Behavior Inventory, PMA—Positive Mental Attitude, PEH—Proper Eating Habits, PA—Preventive Actions, PHA—Prohealth Activities.

**Table 4 healthcare-11-01153-t004:** Associations between life satisfaction, illness acceptance, health behaviors and severity of anxiety and depression symptoms in a group of people with anxiety disorders.

n = 70	AIS	SWLS	HADS-A	HADS-D	HBI	PEH	PA	PMA	PHA
AIS	1.000	−0.014	−0.230	−0.095	0.018	0.084	−0.070	0.004	0.026
SWLS		1.000	−0.274 *	−0.341 *	0.141	0.234	0.082	0.169	−0.099
HADS-A			1.000	0.639 *	−0.239 *	−0.107	−0.153	−0.356 *	−0.309 *
HADS-D				1.000	−0.158	−0.137	−0.122	−0.282 *	−0.069
HBI					1.000	0.727 *	0.744 *	0.776 *	0.567 *
PEH						1.000	0.355 *	0.366 *	0.150
PA							1.000	0.477 *	0.327 *
PMA								1.000	0.525 *
PHA									1.000

AIS—Acceptance of Illness Scale, HADS-A—Hospital Anxiety and Depression Scale—Anxiety, HADS-D—Hospital Anxiety and Depression Scale—Depression, SWLS—The Satisfaction with Life Scale, HBI—Health Behavior Inventory, PMA—Positive Mental Attitude, PEH—Proper Eating Habits, PA—Preventive Actions, PHA—Prohealth Activities. * *p* ≤ 0.05.

## Data Availability

The data presented in this study are available on reasonable request from the corresponding author. The data are not publicly available due to data sensitivity and to protect the interests and privacy of the respondents.

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
