# Peer review of "Health-Promoting Factors and Their Relationships with the Severity of Symptoms in Patients with Anxiety Disorders during the COVID-19 Pandemic"

_healthcare, 2023, doi:10.3390/healthcare11081153_

Round 1

Reviewer 1 Report

One of the limitations of the study is that the patients are already in treatment. It becomes necessary to expose this limitation, which is a factor that can have an impact on the answers in the questionnaires. The authors also did not report the type of treatment the patients are undergoing. It becomes important to inform whether the patients are on some kind of drug (define the class) or only psychotherapy. The lack of information about income is also a weakness because the economic condition of many individuals was quite shaken during the pandemic and this is a factor that can impact on life satisfaction scales.

The covid-19 pandemic was an atypical period that introduced considerable bias into the analysis, making it difficult to understand whether the Health Behavior Inventory and Satisfaction with Life Scale (SWLS) indices were associated with the pandemic or with depression/anxiety status. The lack of a questionnaire about the impact of covid-19 on income, loss of loved ones, illness, impact on work, and state of social isolation at the time of completing the questionnaire weakens the conclusions.

Even with these limitations, the study provides a description of patients with anxiety symptoms during the pandemic assessed using well-chosen scales and therefore has value in itself.

Author Response

Dear Reviewer,

Thank you for your  analysis of our manuscript, we are really obliged. In accordance to your remarks, we have corrected the text as indicated. Below are our answers. On the text, we marked corrected parts in different colour.

We absolutely agree that one of the limitations of the study is that the patients were already in active treatment.

We also did not report the type of treatment the patients are undergoing. According to your suggestion me provided these information into the text in section: limitation of the work.

Additionally we contained in the manuscript other suggested limitation as the lack of information about income, loss of loved ones, illness, impact on work, and state of social isolation at the time of completing the questionnaire and emphasized  weakens the conclusions.

Sincerely, Authors

Reviewer 2 Report

Thank you for the opportunity to review your research. To be considered for publication there needs to be significant amendments made to the Manuscript. Please see my comments below. 

Abstract

1) It is unclear as to how the study was conducted, more information needs to be included regarding the study design and what the study procedures were.

2) Instead of presenting the results in a sequentially manner there needs to be more of a integrated interpretation of your findings.

3) Did the authors analyse gender differences as they said they would?  

4) The material and methods need to be integrated together. It is confusing reading them both separately. 

Introduction

1) The whole introduction is under-referenced, I would like to see more references to support the claims the authors made. 

2) The problem as to why this study is needed must be better defined. We already know that Covid-19 has had a significant impact of people's mental health and quality of life, especially in those with who already had poor mental health. Why is this study needed? What would it add to our current understanding? 

3) If the focus of this study is on individual's with general anxiety disorder there must also be a specific focus on this area too in the introduction. Currently, the scope of this introduction is too broad. 

4) Can the authors separate the introduction into smaller succinct paragraphs? At the moment it is quite challenging to read. 

Methods 

1) The material and methods must include more information other than the outcomes used. More information should be included about the study design, study participants, recruitment procedures, study procedures and participant demographics. 

2) In the statistical analysis paragraph, how did the authors analyse gender differences? Was this conducted? 

3) Similar to the abstract, the material and methods need to be integrated. 

Results 

1) There must be a clearer and more integrated overview of the results presented as opposed to sequentially listing each result. What do these scores indicate? 

2) The analysis of gender differences is included in a table at the end of the manuscript, however, there is no interpretation of this in the results section. 

Discussion

The discussion provides a good interpretation of the study findings in the context of wider literature. I'd like to know how the findings of this research will inform future research and practice. What are the clinical implications of this research? How can we learn from it? 

Conclusion

1) The conclusion needs improving and should not just list the main findings of the study. There must be a meaningful overview of the research. 

Author Response

Dear Reviewer,

Thank you for your thorough analysis of our study, we are very much obliged. In accordance with your review, we have corrected the text as indicated. Below are our answers. On the text, we marked corrected parts on different colour.

Abstract

  1. It is unclear as to how the study was conducted, more information needs to be included regarding the study design and what the study procedures were.

We changed the material and methods in the abstract.

2) Instead of presenting the results in a sequentially manner there needs to be more of a integrated interpretation of your findings.

We change that in the abstract.

3) Did the authors analyse gender differences as they said they would? 

We did analyze gender difference, but there were mostly no statistical significances. We would like to enlarge the research group and then make some gender analyzes again. 

4) The material and methods need to be integrated together. It is confusing reading them both separately. 

We did that, thank you.

Introduction

  1. The whole introduction is under-referenced, I would like to see more references to support the claims the authors made. 

We changed that and added more references, with the critical view.

2) The problem as to why this study is needed must be better defined. We already know that Covid-19 has had a significant impact of people's mental health and quality of life, especially in those with who already had poor mental health. Why is this study needed? What would it add to our current understanding? 

We made some statements in the introduction, as you suggested.

3) If the focus of this study is on individual's with general anxiety disorder there must also be a specific focus on this area too in the introduction. Currently, the scope of this introduction is too broad. 

We made some changes in the introduction, as you suggested.

4) Can the authors separate the introduction into smaller succinct paragraphs? At the moment it is quite challenging to read. 

We made two paragraphs in the introduction - the „life satisfaction” section and the „impact of COVID-19 on patients with anxiety disorder”.

Methods 

  1. The material and methods must include more information other than the outcomes used. More information should be included about the study design, study participants, recruitment procedures, study procedures and participant demographics. 

We precisized the material in the text, as you suggested.

2) In the statistical analysis paragraph, how did the authors analyse gender differences? Was this conducted? 

We did analyze gender difference, but there were mostly no statistical significances. We would like to enlarge the research and then make some gender analyses again. 

3) Similar to the abstract, the material and methods need to be integrated. 

We did that, thank you.

Results 

  1. There must be a clearer and more integrated overview of the results presented as opposed to sequentially listing each result. What do these scores indicate? 

We included theoretical components of each scales we used and compare them to our results from the study.

2) The analysis of gender differences is included in a table at the end of the manuscript, however, there is no interpretation of this in the results section. 

We add interpretation in the results about it, thank you for your suggestion.

Discussion

The discussion provides a good interpretation of the study findings in the context of wider literature. I'd like to know how the findings of this research will inform future research and practice. What are the clinical implications of this research? How can we learn from it? 

When designing our study, our goal was to measure the relationship between health-promoting behaviors and their reliable use in the prevention of anxiety and depressive disorders. Bearing in mind the fact that the COVID-19 pandemic was very degrading in terms of the quality of life for all people, regardless of the country they live in, it was crucial to find practices that could minimize not only the reduction in the quality of life, but also prevent the aggravation of anxiety disorders (both existing - as in our study, in people diagnosed with GAD and newly formed) and depressive disorders. Pro-health practices seem to be still underestimated as a non-pharmacological prophylactic method. We think that emphasizing the practical importance of pro-health practices - both Positive Mental Attitude (PMA) (avoiding strong emotions, tension, and stressful and depressing situations), Proper Eating Habits (PEH) (type and frequency of food consumed), Preventive Actions (PA) (following health recommendations, obtaining information about health and illness), and Pro-health Activities (PhA) (everyday habits: sleep, recreation, physical activity) is very important, what is more, it should initiate an information campaign among doctors - not only psychiatrists who treat mental disorders on a daily basis, but also family doctors/GPs. What's more, further research can focus on which of these health-promoting practices (which activities or actions, exactly) are most important in reducing the sense of anxiety or depressive disorders among people.

Conclusion

  1. The conclusion needs improving and should not just list the main findings of the study. There must be a meaningful overview of the research. 

We made changes in conclusions.

Authors

Round 2

Reviewer 2 Report

Thank you, you have responded to my comments well and I believe the manuscript has been improved.